# Characterizing a thermostable Cas9 for bacterial genome editing and silencing

Ioannis Mougiakos[1], Prarthana Mohanraju [1], Elleke F. Bosma[1,3], Valentijn Vrouwe[1], Max Finger Bou[1], Mihris I.S. Naduthodi[1], Alex Gussak[1], Rudolf B.L. Brinkman[2], Richard van Kranenburg [1,2] & John van der Oost[1]

CRISPR-Cas9-based genome engineering tools have revolutionized fundamental research and biotechnological exploitation of both eukaryotes and prokaryotes. However, the mesophilic nature of the established Cas9 systems does not allow for applications that require enhanced stability, including engineering at elevated temperatures. Here we identify and characterize ThermoCas9 from the thermophilic bacterium *Geobacillus thermodenitrificans* T12. We show that in vitro ThermoCas9 is active between 20 and 70 °C, has stringent PAM-preference at lower temperatures, tolerates fewer spacer-protospacer mismatches than SpCas9 and its activity at elevated temperatures depends on the sgRNA-structure. We develop ThermoCas9-based engineering tools for gene deletion and transcriptional silencing at 55 °C in *Bacillus smithii* and for gene deletion at 37 °C in *Pseudomonas putida*. Altogether, our findings provide fundamental insights into a thermophilic CRISPR-Cas family member and establish a Cas9-based bacterial genome editing and silencing tool with a broad temperature range.

[1] Laboratory of Microbiology, Wageningen University, Stippeneng 4, 6708 WE Wageningen, The Netherlands. [2] Corbion, Arkelsedijk 46, 4206 AC Gorinchem, The Netherlands. [3] Present address: The Novo Nordisk Foundation Center for Biosustainability, Technical University of Denmark, Kemitorvet B220, 2800 Kgs Lyngby, Denmark. Ioannis Mougiakos, Prarthana Mohanraju and Elleke F. Bosma contributed equally to this work. Correspondence and requests for materials should be addressed to J.v.d.O. (email: john.vanderoost@wur.nl)

Clustered regularly interspaced short palindromic repeats (CRISPR) and the CRISPR-associated (Cas) proteins provide adaptive and heritable immunity in prokaryotes against invading genetic elements[1–4]. CRISPR-Cas systems are subdivided into two classes (1 and 2) and six types (I–VI), depending on their complexity and signature proteins[5]. Class 2 systems, including type-II CRISPR-Cas9 and type V CRISPR-Cas12a (previously called CRISPR- Cpf1) have recently been exploited as genome engineering tools for both eukaryotes[6–10] and prokaryotes[11–13]. These systems are among the simplest CRISPR-Cas systems known as they introduce targeted double-stranded DNA breaks (DSBs) based on a ribonucleoprotein (RNP) complex formed by a single Cas endonuclease and an RNA guide.

The guide of Cas9 consists of a crRNA (CRISPR RNA): tracrRNA (trans-activating-CRISPR-RNA) duplex. For engineering purposes, the crRNA:tracrRNA duplex has been simplified by generating a chimeric, single guide RNA (sgRNA) to guide Cas9 upon co-expression[14]. In addition, cleavage of the target DNA requires a protospacer adjacent motif (PAM): a 3–8 nucleotide (nt) long sequence located next to the targeted protospacer that is highly variable between different Cas9 proteins[15–17]. Cas9 endonucleases contain two catalytic domains, denoted as RuvC and HNH. Substituting catalytic residues in one of these domains results in Cas9 nickase variants, and in both domains in an inactive variant[18–20]. The inactive or dead Cas9 (dCas9) has been instrumental as an efficient gene silencing system and for modulating the expression of essential genes[11,21,22].

To date, *Streptococcus pyogenes* Cas9 (SpCas9) is the best characterized and most widely employed Cas9 for genome engineering. Although a few other type-II systems have been exploited for bacterial genome engineering purposes, none of them is derived from a thermophilic organism[23]. Characterization of such

CRISPR-Cas systems would be interesting to gain fundamental insights as well as to develop novel applications.

Although basic genetic tools are available for a number of thermophiles[24–27], the efficiency of these tools is still too low to enable full exploration and exploitation of this interesting group of organisms. Based on our finding that SpCas9 is not active in vivo at or above 42 °C, we have previously developed a SpCas9-based engineering tool for facultative thermophiles, combining homologous recombination at elevated temperatures and SpCas9-based counter-selection at moderate temperatures[28]. However, a Cas9-based editing and silencing tool for obligate thermophiles is not yet available as SpCas9 is not active at elevated temperatures[28,29], and to date no thermophilic Cas9 has been adapted for such purpose. Here we describe the characterization of ThermoCas9: an RNA-guided DNA-endonuclease from the CRISPR-Cas type-IIC system of the thermophilic bacterium *Geobacillus thermodenitrificans* T12[30]. We show that ThermoCas9 is active in vitro between 20 and 70 °C and demonstrate the effect of the sgRNA-structure on its thermostability. We apply ThermoCas9 for in vivo genome editing and silencing of the industrially important thermophile *Bacillus smithii* ET 138[31] at 55 °C, creating the first, to our knowledge, Cas9-based genome engineering tool readily applicable to thermophiles. In addition, we apply ThermoCas9 for in vivo genome editing of the mesophile *Pseudomonas putida* KT2440, for which to date no CRISPR-Cas9-based editing tool had been described[32,33], confirming the wide temperature range and broad applicability of this novel Cas9 system.

## Results

**ThermoCas9 identification and purification.** We recently isolated and sequenced *Geobacillus thermodenitrificans* strain T12, a Gram positive, moderately thermophilic bacterium with an

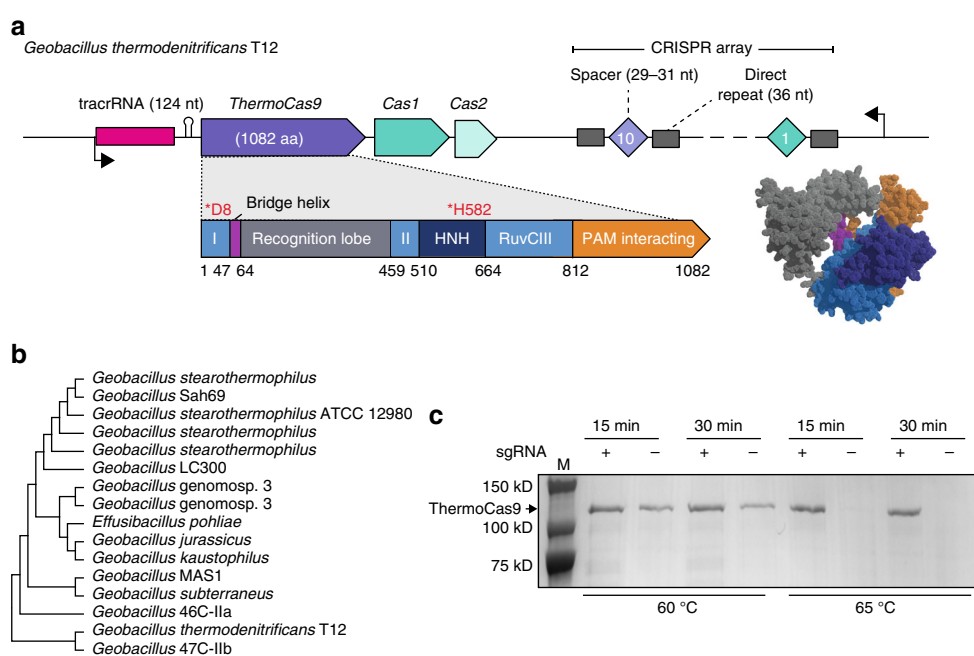

**Fig. 1** The *Geobacillus thermodenitrificans* T12 type-IIC CRISPR-Cas locus encoding ThermoCas9. **a** Schematic representation of the genomic locus encoding ThermoCas9. The domain architecture of ThermoCas9 based on sequence comparison, with predicted active sites residues highlighted in magenta. A homology model of ThermoCas9 generated using Phyre 2[57] is shown, with different colors for the domains. **b** Phylogenetic tree of Cas9 orthologues highly identical to ThermoCas9. Evolutionary analysis was conducted in MEGA7[58]. **c** SDS-PAGE of ThermoCas9 after purification by metal-affinity chromatography and gel filtration. The migration of the obtained single band is consistent with the theoretical molecular weight of 126 kD of the apo-ThermoCas9

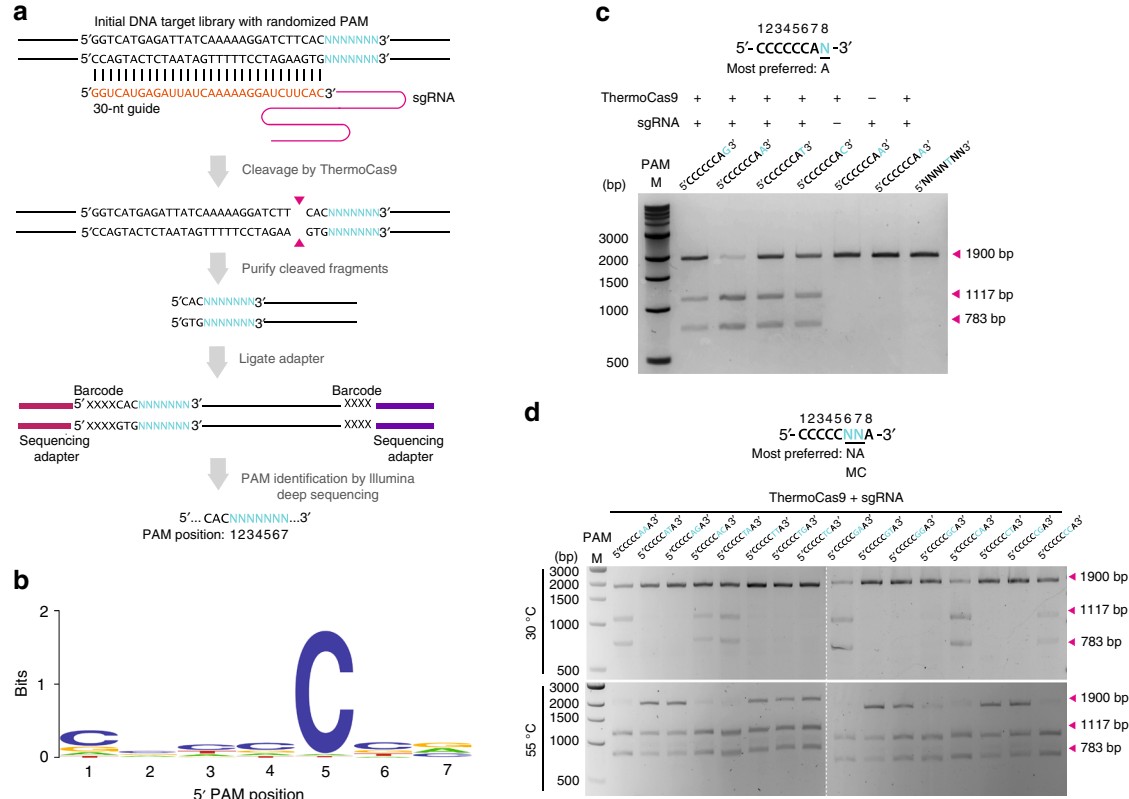

**Fig. 2** ThermoCas9 PAM analysis. **a** Schematic illustrating the in vitro cleavage assay for discovering the position and identity (5'-NNNNNNN-3') of the protospacer adjacent motif (PAM). Magenta triangles indicate the cleavage position. **b** Sequence logo of the consensus 7 nt long PAM of ThermoCas9, obtained by comparative analysis of the ThermoCas9-based cleavage of target libraries. Letter height at each position is measured by information content. **c** Extension of the PAM identity to the eighth position by in vitro cleavage assay. Four linearized plasmid targets, each containing a distinct 5'-CCCCCCAN-3' PAM, were incubated with ThermoCas9 and sgRNA at 55 °C for 1 h, then analyzed by agarose gel electrophoresis. Supplementary Fig. 8 shows the uncropped gel image. **d** In vitro cleavage assays for DNA targets with different PAMs at 30 and 55 °C. Sixteen linearized plasmid targets, each containing one distinct 5'-CCCCCNNA-3' PAM, were incubated with ThermoCas9 and sgRNA, then analyzed for cleavage efficiency by agarose gel electrophoresis. See also Supplementary Fig. 3. Supplementary Fig. 9 shows the uncropped gel images

optimal growth temperature at 65 °C[30]. Contrary to previous claims that type-II CRISPR-Cas systems are not present in thermophilic bacteria[34], the sequencing results revealed the existence of a type-IIC CRISPR-Cas system in the genome of *G. thermodenitrificans* T12 (Fig. 1a)[35]. The Cas9 endonuclease of this system (ThermoCas9) was predicted to be relatively small (1082 amino acids) compared to other Cas9 orthologues, such as SpCas9 (1368 amino acids). The size difference is mostly due to a truncated REC lobe, as has been demonstrated for other small Cas9 orthologues (Supplementary Fig. 1)[36]. Furthermore, ThermoCas9 was expected to be active at least around the temperature optimum of *G. thermodenitrificans* T12[30]. Using the ThermoCas9 sequence as query, we performed BLAST-P searches in the NCBI/non-redundant protein sequences dataset, and found a number of highly identical Cas9 orthologues (87–99% identity at amino acid level, Supplementary Table 1), mostly within the *Geobacillus* genus, supporting the idea that ThermoCas9 is part of a highly conserved defense system of thermophilic bacteria (Fig. 1b). These characteristics suggested it may be a potential candidate for exploitation as a genome editing and silencing tool for thermophilic microorganisms, and for conditions at which enhanced protein robustness is required.

We initially performed in silico prediction of the crRNA and tracrRNA modules of the *G. thermodenitrificans* T12 CRISPR-Cas system using a previously described approach[11,36]. Based on this prediction, a 190 nt sgRNA chimera was designed by linking the predicted full-size crRNA (30 nt long spacer followed by 36 nt

long repeat) and tracrRNA (36 nt long anti-repeat followed by a 88 nt sequence with three predicted hairpin structures). ThermoCas9 was heterologously expressed in *E. coli* and purified to homogeneity. Hypothesizing that the loading of the sgRNA to the ThermoCas9 would stabilize the protein, we incubated purified apo-ThermoCas9 and ThermoCas9 loaded with in vitro transcribed sgRNA at 60 °C and 65 °C, for 15 and 30 min. SDS-PAGE analysis showed that the purified ThermoCas9 denatures at 65 °C but not at 60 °C, while the denaturation temperature of ThermoCas9-sgRNA complex is above 65 °C (Fig. 1c). The demonstrated thermostability of ThermoCas9 implied its potential as a thermo-tolerant CRISPR-Cas9 genome editing tool, and encouraged us to analyze some relevant molecular features in more detail.

**ThermoCas9 PAM determination.** The first step towards the characterization of ThermoCas9 was the in silico prediction of its PAM preferences for successful cleavage of a DNA target. We used the 10 spacers of the *G. thermodenitrificans* T12 CRISPR locus to search for potential protospacers in viral and plasmid sequences using CRISPRtarget[37]. As only two hits were obtained with phage genomes (Supplementary Fig. 2A), it was decided to proceed with an in vitro PAM determination approach. The predicted sgRNA sequence was generated by in vitro transcription, including a spacer that should allow for ThermoCas9-based targeting of linear dsDNA substrates with a matching protospacer. The protospacer was flanked at its 3'-end by randomized 7-base pair (bp)

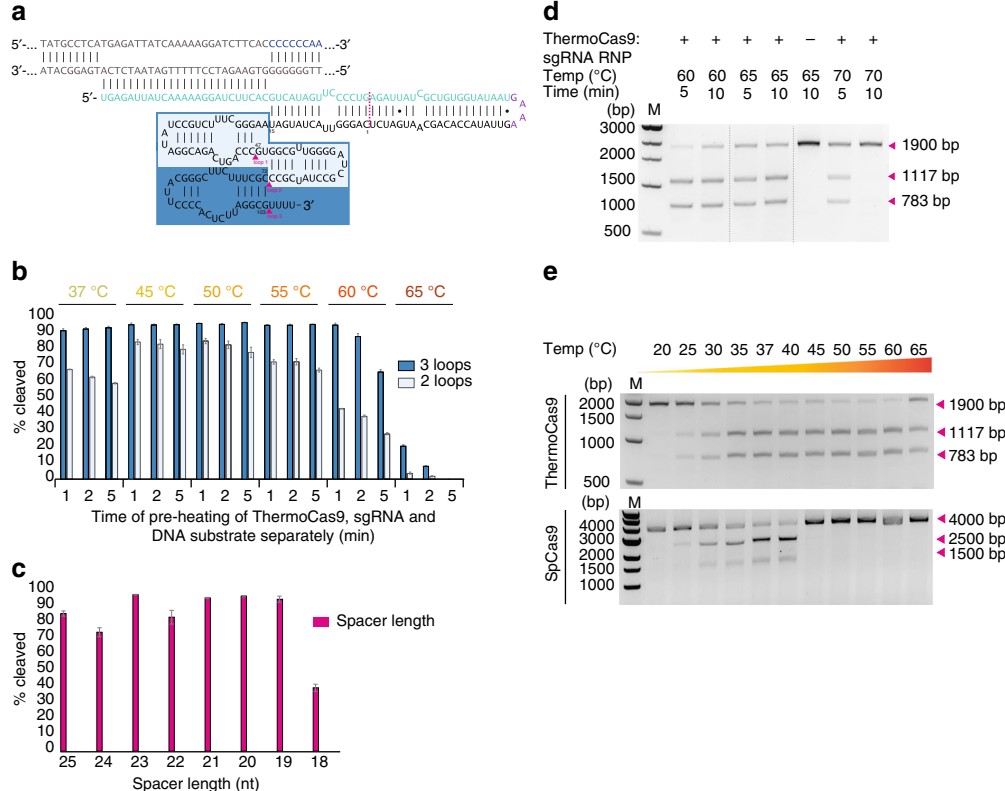

**Fig. 3** ThermoCas9 temperature range and effect of sgRNA-binding. **a** Schematic representation of the sgRNA and a matching target DNA. The target DNA, the PAM and the crRNA are shown in gray, blue and turquoise, respectively. The site where the crRNA is linked with the tracrRNA is shown in purple. The dark blue and light blue boxes indicate the predicted three and two loops of the tracrRNA, respectively. The 41-nt truncation of the repeat-anti-repeat region and the three loops of the sgRNA are indicated by the magenta dotted line and magenta triangles, respectively. **b** The importance of the predicted three stem-loops of the tracrRNA scaffold was tested by transcribing truncated variants of the sgRNA and evaluating their ability to guide ThermoCas9 to cleave target DNA at various temperatures. Average values of three replicates are shown, with error bars representing S.D. The blots of one of the replicates are shown in Supplementary Fig. 10. **c** The importance of the length of the spacer was tested by transcribing truncated variants of the initial spacer in the sgRNA and evaluating their ability to guide ThermoCas9 to cleave target DNA at 55 °C. Average values of three replicates are shown, with error bars representing S. D.. The blots of one of the replicates are shown in Supplementary Fig. 11. **d** To identify the maximum temperature, endonuclease activity of ThermoCas9: sgRNA RNP complex was assayed after incubation at 60, 65, and 70 °C for 5 or 10 min. The pre-heated DNA substrate was added and the reaction was incubated for 1 h at the corresponding temperature. **e** Comparison of active temperature range of ThermoCas9 and SpCas9 by activity assays conducted after 5 min of incubation at the indicated temperature. The pre-heated DNA substrate was added and the reaction was incubated for 1 h at the same temperature

sequences. After performing ThermoCas9-based cleavage assays at 55 °C, the cleaved sequences of the library (together with a non-targeted library sample as control) were separated from uncleaved sequences, by gel electrophoresis, and analyzed by deep-sequencing in order to identify the ThermoCas9 PAM preference (Fig. 2a). The sequencing results revealed that Thermo-Cas9 introduces double-stranded DNA breaks that, in analogy with the mesophilic Cas9 variants, are located mostly between the third and the forth PAM proximal nucleotides, at the 3′ end of the protospacer. Moreover, the cleaved sequences revealed that ThermoCas9 recognizes a 5′-NNNNCNR-3′ PAM, with subtle preference for cytosine at the first, third, forth, and sixth PAM positions (Fig. 2b). Recent studies have revealed the importance of the eighth PAM position for target recognition of some Type-IIC Cas9 orthologues[17,38]. For this purpose, and taking into account the results from the in silico ThermoCas9 PAM prediction (Supplementary Fig. 2), we performed additional PAM determination assays. This revealed optimal targeting efficiency in the presence of an adenine at the eighth PAM position (Fig. 2c). Interestingly, despite the limited number of hits, the aforementioned in silico PAM prediction (Supplementary Fig. 2B) also suggested the significance of a cytosine at the fifth and an adenine at the eighth PAM positions.

To further clarify the ambiguity of the PAM at the sixth and seventh PAM positions, we generated a set of 16 different target DNA fragments in which the matching protospacer was flanked by 5′-CCCCCNNA-3′ PAMs. Cleavage assays of these fragments (each with a unique combination of the sixth and seventh nucleotide) were performed in which the different components (ThermoCas9, sgRNA guide, dsDNA target) were pre-heated separately at different temperatures (20, 30, 37, 45, 55, and 60 °C) for 10 min before combining and incubating them for 1 h at the corresponding assay temperature. When the assays were performed at temperatures between 37 and 60 °C, all the different DNA substrates were cleaved (Fig. 2d, S3). However, the most digested target fragments consisted of PAM sequences (fifth to eighth PAM positions) 5′-CNAA-3′ and 5′-CMCA-3′, whereas the least digested targets contained a 5′-CAKA-3′ PAM. At 30 °C, only cleavage of the DNA substrates with the optimal PAM sequences (fifth to eighth PAM positions) 5′-CNAA-3′ and 5′-CMCA-3′ was observed (Fig. 2d). Finally, at 20 °C only the DNA substrates with (fifth to eighth PAM positions) 5′-CVAA-3′ and 5′-CCCA-3′ PAM sequences were targeted (Supplementary Fig. 3), making these sequences the most preferred PAMs. Our findings demonstrate that at its lower temperature limit, ThermoCas9 only cleaves fragments with a preferred PAM.

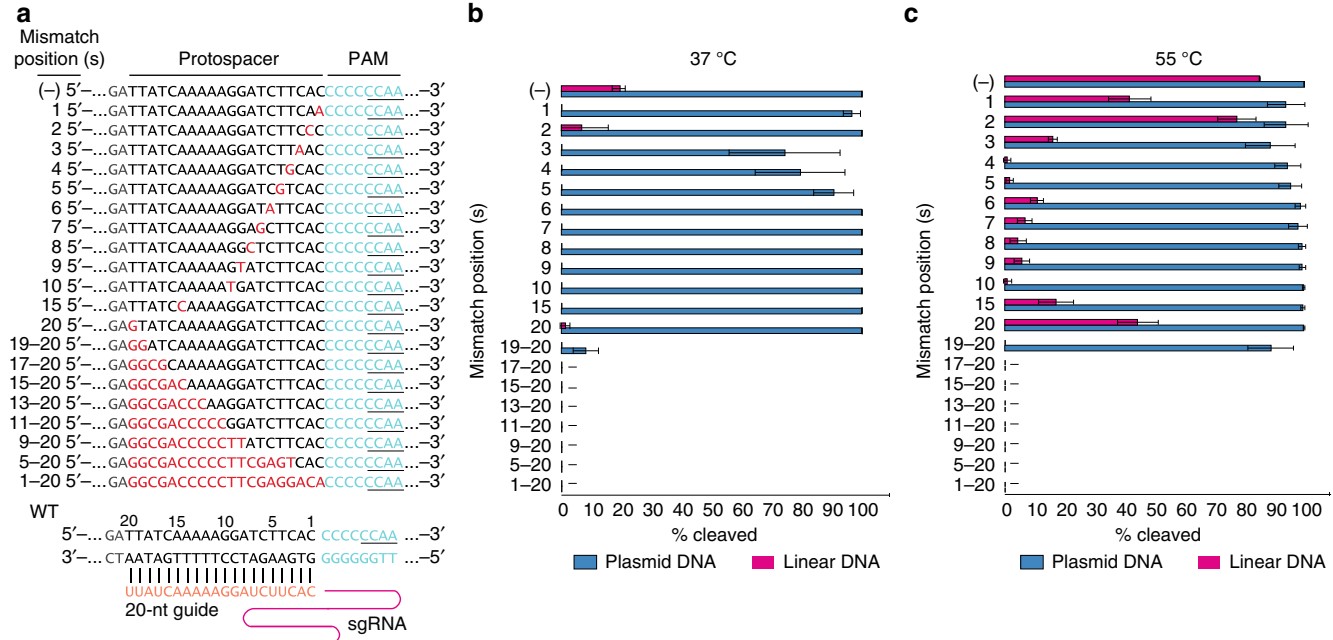

**Fig. 4** Protospacer targeting Specificity of ThermoCas9. **a** Scheme of the generated mismatch protospacers library, employed for evaluating the ThermoCas9:sgRNA targeting specificity in vitro. The mismatch spacer-protospacer positions are shown in red, the PAM in light blue with the fifth to eighth positions underlined. **b** Graphical representation of the ThermoCas9:sgRNA cleavage efficiency over linear or plasmid targets with different mismatches at 37 °C. The percentage of cleavage was calculated based on integrated band intensities after gel electrophoresis. Average values from three biological replicates are shown, with error bars representing S.D. The blots of one of the replicates are shown in Supplementary Fig. 12. **c** Graphical representation of the ThermoCas9:sgRNA cleavage efficiency over linear or plasmid targets with different mismatches at 55 °C. The percentage of cleavage was calculated based on integrated band intensities after gel electrophoresis. Average values from three biological replicates are shown, with error bars representing S.D. The blots of one of the replicates are shown in Supplementary Fig. 12

This characteristic could be exploited during in vivo editing processes, for example to avoid off-target effects.

**Metal ion dependency**. Previously characterized, mesophilic Cas9 endonucleases employ divalent cations to catalyze the generation of DSBs in target DNA[14,39]. To determine the ion dependency of ThermoCas9 cleavage activity, plasmid cleavage assays were performed in the presence of one of the following divalent cations: $Mg^{2+}$, $Ca^{2+}$, $Mn^{2+}$, $Fe^{2+}$, $Co^{2+}$, $Ni^{2+}$, and $Zn^{2+}$; an assay with the cation-chelating agent EDTA was included as negative control. As expected, target dsDNA was cleaved in the presence of divalent cations and remained intact in the presence of EDTA (Supplementary Fig. 5A). The DNA cleavage activity of ThermoCas9 was the highest when $Mg^{2+}$ and $Mn^{2+}$ was added to the reaction consistent with other Cas9 variants[14,20,40]. Addition of $Fe^{2+}$, $Co^{2+}$, $Ni^{2+}$, or $Zn^{2+}$ ions also mediated cleavage. $Ca^{2+}$ only supported plasmid nicking, suggesting that with this cation only one of the endonuclease domains is functional.

**Thermostability and truncations**. The predicted tracrRNA consists of the anti-repeat region followed by three hairpin structures (Fig. 3a). Using the tracrRNA along with the crRNA to form a sgRNA chimera resulted in successful guided cleavage of the DNA substrate. It was observed that a 41-nt long deletion of the spacer distal end of the full-length repeat-anti-repeat hairpin (Fig. 3a), most likely better resembling the dual guide's native state, had little to no effect on the DNA cleavage efficiency. The effect of further truncation of the predicted hairpins (Fig. 3a) on the cleavage efficiency of ThermoCas9 was evaluated by performing a cleavage time-series in which all the components (sgRNA, ThermoCas9, substrate DNA) were pre-heated

separately at different temperatures (37–65 °C) for 1, 2 and 5 min before combining and incubating them for 1 h at various assay temperatures (37–65 °C). The number of predicted stem-loops of the tracrRNA scaffold seemed to play a crucial role in DNA cleavage; when all three loops were present, the cleavage efficiency was the highest at all tested temperatures, whereas the efficiency decreased upon removal of the 3′ hairpin (Fig. 3b). Moreover, the cleavage efficiency drastically dropped upon removal of both the middle and the 3′ hairpins (Supplementary Fig. 4). Whereas pre-heating ThermoCas9 at 65 °C for 1 or 2 min resulted in detectable cleavage, the cleavage activity was abolished after 5 min incubation. The thermostability assay showed that sgRNA variants without the 3′stem-loop result in decreased stability of the ThermoCas9 protein at 65 °C, indicating that a full length tracrRNA is required for optimal ThermoCas9-based DNA cleavage at elevated temperatures. Additionally, we also varied the lengths of the spacer sequence (from 25 to 18 nt) and found that spacer lengths of 23, 21, 20, and 19 cleaved the targets with the highest efficiency. The cleavage efficiency drops significantly when a spacer of 18 nt is used (Fig. 3c).

In vivo, the ThermoCas9:sgRNA RNP complex is probably formed within minutes. Together with the above findings, this motivated us to evaluate the activity and thermostability of the RNP. Pre-assembled RNP complex was heated at 60, 65, and 70 °C for 5 and 10 min before adding pre-heated DNA and subsequent incubation for 1 h at 60, 65, and 70 °C. Strikingly, we observed that the ThermoCas9 RNP was active up to 70 °C, in spite of its pre-heating for 5 min at 70 °C (Fig. 3d). This finding confirmed our assumption that the ThermoCas9 stability strongly correlates with the association of an appropriate sgRNA guide[41].

Proteins of thermophilic origin generally retain activity at lower temperatures. Hence, we set out to compare the

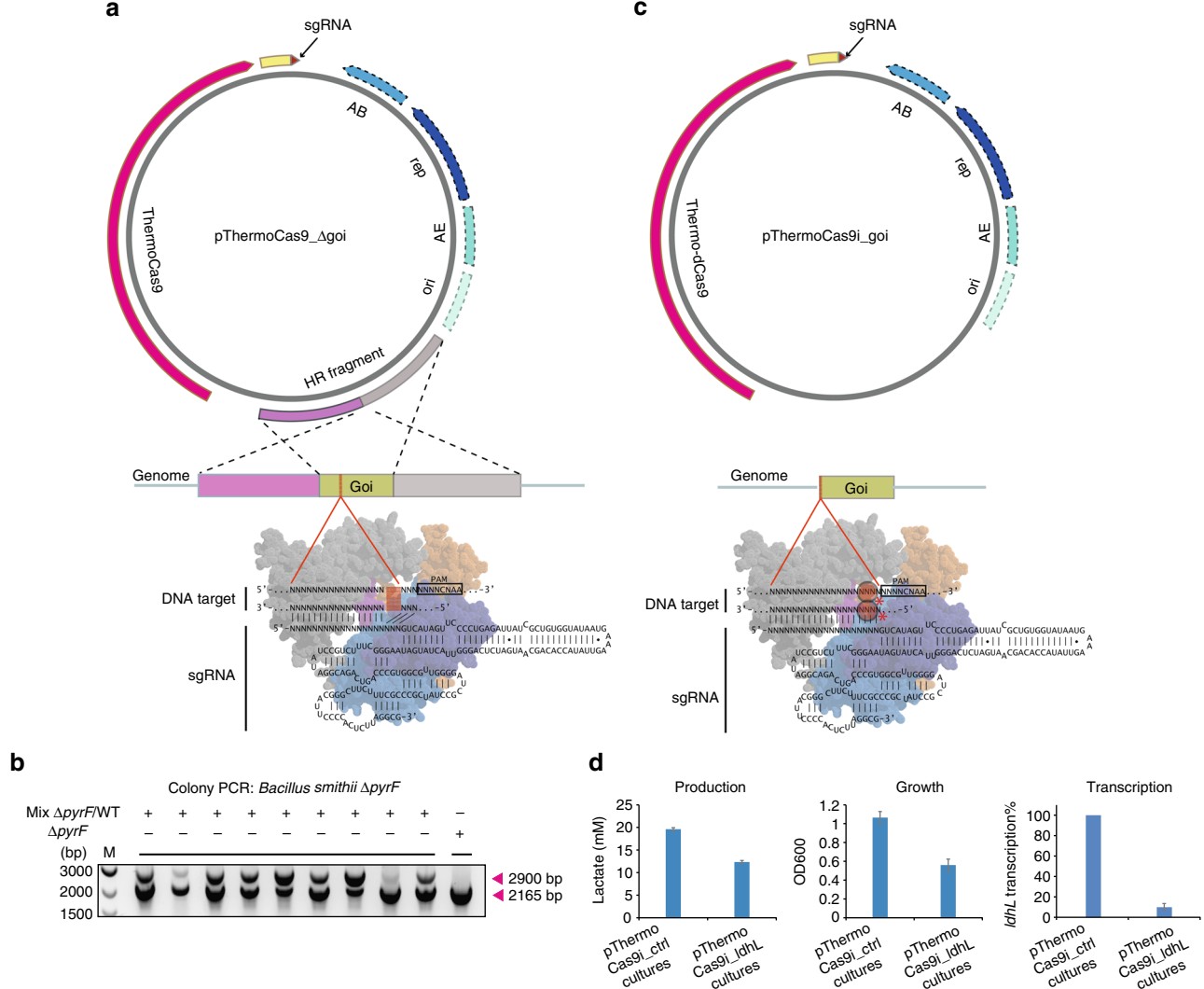

**Fig. 5** ThermoCas9-based genome engineering in prokaryotes. **a** Schematic overview of the basic pThermoCas9_Δgene-of-interest (goi) construct. The *thermocas9* gene was introduced either to the pNW33n (*B. smithii*) or to the pEMG (*P. putida*) vector. Homologous recombination flanks were introduced upstream *thermocas9* and encompassed the 1 kb (*B. smithii*) or 0.5 kb (*P. putida*) upstream and 1 kb or 0.5 kb downstream region of the gene of interest (goi) in the targeted genome. A sgRNA-expressing module was introduced downstream the *thermocas9* gene. As the origin of replication (ori), replication protein (rep), antibiotic resistance marker (AB) and possible accesory elements (AE) are backbone specific, they are represented with dotted outline. **b** Agarose gel electrophoresis showing the resulting products from genome-specific PCR on ten colonies from the ThermoCas9-based *pyrF* deletion process from the genome of *B. smithii* ET 138. All ten colonies contained the Δ*pyrF* genotype and one colony was a clean Δ*pyrF* mutant, lacking the wild-type product. Supplementary Fig. 13 shows the uncropped gel image. **c** Schematic overview of the basic pThermoCas9i_goi construct. Aiming for the expression of a catalytically inactive ThermoCas9 (ThermodCas9: D8A, H582A mutant), the corresponding mutations were introduced to create the *thermodcas9* gene. The *thermodcas9* gene was introduced to the pNW33n vector. A sgRNA-expressing module was introduced downstream the *thermodcas9*. **d** Graphical representation of the production, growth and RT-qPCR results from the *ldhL* silencing experiment using ThermodCas9. The graphs represent the lactate production, optical density at 600 nm and percentage of *ldhL* transcription in the repressed cultures compared to the control cultures. Average values from three biological replicates are shown, with error bars representing S.D

ThermoCas9 temperature range to that of the *Streptococcus pyogenes* Cas9 (SpCas9). Both Cas9 homologues were subjected to in vitro activity assays between 20 and 65 °C. Both proteins were incubated for 5 min at the corresponding assay temperature prior to the addition of the sgRNA and the target DNA molecules. In agreement with previous analysis[28,29], the mesophilic SpCas9 was active only between 25 and 44 °C (Fig. 3e); above this temperature SpCas9 activity rapidly decreased to undetectable levels. In contrast, ThermoCas9 cleavage activity could be detected between 25 and 65 °C (Fig. 3). This indicates the potential to use ThermoCas9 as a genome editing tool for both thermophilic and mesophilic organisms.

Based on previous reports that certain type-IIC systems were efficient single stranded DNA cutters[40,41], we tested the activity of ThermoCas9 on ssDNA substrates. However, no cleavage was observed, indicating that ThermoCas9 is a dsDNA nuclease (Supplementary Fig. 5B).

**Spacer-protospacer mismatch tolerance.** The targeting specificity and spacer-protospacer mismatch tolerance of a Cas9 endonuclease provide vital information for the development of the Cas9 into a genome engineering tool. To investigate the targeting specificity of ThermoCas9 toward a selected protospacer,

we constructed a target plasmid library by introducing either single-mismatches or multiple-mismatches to the previously employed protospacer (Fig. 4a). Each member of the plasmid library, and its PCR-linearized derivative, was separately used as substrate for in vitro ThermoCas9 cleavage assays in three independent experiments, both at 37 and 55 °C.

The ThermoCas9:sgRNA activity on linear dsDNA targets was abolished at 37 °C for most of the single-mismatch targets (Fig. 4b). Noteworthy exceptions were the targets with single-mismatches at the PAM proximal position 2 and PAM distal position 20, which allowed for weak cleavage (Fig. 4b). At 55 °C, the cleavage efficiency for single-mismatch linear targets was higher than at 37 °C; however, it was strongly hampered for most of the tested targets, especially for single-mismatches at the PAM proximal positions 4, 5, and 10 (Fig. 4c). On the contrary, single-mismatches at positions 1, 2, and 20 were the most tolerated for cleavage (Fig. 4c).

In complete contrast to the linear targets with single-mismatches, all the corresponding plasmid targets were cleaved by the ThermoCas9:sgRNA complex at 37 °C, regardless the position of the mismatch, with preference for the targets with single-mismatches at the PAM proximal positions 2, 6–10, 15, and 20 (Fig. 4b). At 55 °C, a similar trend was observed, with elevated cleavage rates (Fig. 4c). Remarkably, the ThermoCas9: sgRNA activity was impeded for both linear and plasmid targets with multiple-mismatches as there was no detectable cleavage for most of these targets at the tested temperatures (Fig. 4b, c). Notable exception was the target with a double mismatch at positions 19 and 20 which was cleaved at both tested temperatures but more prominently at 55 °C (Fig. 4b, c).

**ThermoCas9-based gene deletion in the thermophile B. smithii.** We set out to develop a ThermoCas9-based genome editing tool for thermophilic bacteria. This group of bacteria is of great interest both from a fundamental as well as from an applied perspective. For biotechnological applications, their thermophilic nature results in for example less cooling costs, higher reaction rates and less contamination risk compared to the widely used mesophilic industrial work horses such as E. coli[24,25,42,43]. Here we show a proof of principle study on the use of ThermoCas9 as genome editing tool for thermophiles, employing B. smithii ET 138 cultured at 55 °C. Its wide substrate utilization range, thermophilic and facultative anaerobic nature, combined with its genetic amenability make this an organism with high potential as platform organism for the production of green chemicals in a biorefinery[24,28,31,44]. In order to use a minimum of genetic parts, we followed a single plasmid approach. We constructed a set of pNW33n-based pThermoCas9 plasmids containing the thermocas9 gene under the control of the native xylL promoter (P$_{xylL}$), a homologous recombination template for repairing Cas9-induced double-stranded DNA breaks within a gene-of-interest, and a sgRNA expressing module under control of the constitutive pta promoter (P$_{pta}$) from Bacillus coagulans (Fig. 5a).

The first goal was the deletion of the full length pyrF gene from the genome of B. smithii ET 138. The pNW33n-derived plasmids pThermoCas9_bsΔpyrF1 and pThermoCas9_bsΔpyrF2 were used for expression of different ThermoCas9 guides with spacers targeting different sites of the pyrF gene, while a third plasmid (pThermoCas9_ctrl) contained a random non-targeting spacer in the sgRNA expressing module. Transformation of B. smithii ET 138 competent cells at 55 °C with the control plasmids pNW33n (no guide) and pThermoCas9_ctrl resulted in the formation of ~200 colonies each. Out of 10 screened pThermoCas9_ctrl colonies, none contained the ΔpyrF genotype, confirming findings from previous studies that, in the absence of appropriate counter-selection, homologous recombination in B. smithii ET 138 is not sufficient to obtain clean mutants[28,44]. In contrast, transformation with the pThermoCas9_bsΔpyrF1 and pThermoCas9_bsΔpyrF2 plasmids resulted in 20 and 0 colonies, respectively. Out of the ten pThermoCas9_ΔpyrF1 colonies screened, one was a clean ΔpyrF mutant whereas the rest had a mixed wild-type/ΔpyrF genotype (Fig. 5b), proving the applicability of the system, as the designed homology directed repair of the targeted pyrF gene was successful. Contrary to eukaryotes, most prokaryotes including B. smithii do not possess a functional NHEJ system, and hence DSBs induced by Cas9 have been shown to be lethal in the absence of a functional HR system and/or of an appropriate HR template[11,28]. Hence, Cas9 functions as stringent counter-selection system to kill cells that have not performed the desired HR prior to Cas9 cleavage[11,28,45]. The combination of lack of NHEJ and low HR-frequencies found in most prokaryotes provides the basis for the power of Cas9-based editing but also creates the need for tight control of Cas9 activity[11,28,45]. As the promoter we use here for thermocas9 expression is not sufficiently controllable and HR is inefficient in B. smithii[28,44], the low number (pyrF1) or even complete lack (pyrF2) of colonies we observed here in the presence of an HR template confirms the high in vivo activity of ThermoCas9 at 55 °C. In the SpCas9-based counter-selection system, we previously developed for B. smithii, the activity of Cas9 was very tightly controlled by the growth temperature rather than by gene expression. This allowed for extended time for the cells to perform HR prior to Cas9 counter-selection, resulting in a higher pyrF deletion efficiency[28]. We anticipate that the use of a tightly controlled promoter will increase efficiencies of the ThermoCas9-system.

**ThermoCas9-based gene deletion in the mesophile P. putida.** To broaden the applicability of the ThermoCas9-based genome editing tool and to evaluate whether our in vitro results could be confirmed in vivo, we next evaluated its activity in the mesophilic Gram-negative bacterium P. putida KT2440. This soil bacterium is well-known for its unusual metabolism and biodegradation capacities, especially of aromatic compounds. Recently, interest in this organism has further increased due to its potential as platform host for biotechnology purposes using metabolic engineering[46,47]. However, to date no CRISPR-Cas-based editing system has been reported for P. putida whereas such a system would greatly increase engineering efficiencies and enhance further study and use of this organism[32,33]. Once more, we followed a single plasmid approach and combined homologous recombination and ThermoCas9-based counter-selection. We constructed the pEMG-based pThermoCas9_ppΔpyrF plasmid containing the thermocas9 gene under the control of the 3-methylbenzoate-inducible Pm-promoter, a homologous recombination template for deletion of the pyrF gene and a sgRNA expressing module under the control of the constitutive P3 promoter. After transformation of P. putida KT2440 cells and PCR confirmation of plasmid integration, a colony was inoculated in selective liquid medium for overnight culturing at 37 °C. The overnight culture was used for inoculation of selective medium and ThermoCas9 expression was induced with 3-methylbenzoate. Subsequently, dilutions were plated on non-selective medium, supplemented with 3-methylbenzoate. For comparison, we performed a parallel experiment without inducing ThermoCas9 expression with 3-methylbenzoate. The process resulted in 76 colonies for the induced culture and 52 colonies for the non-induced control culture. For the induced culture, 38 colonies (50%) had a clean deletion genotype and 6 colonies had mixed wild-type/deletion genotype. On the contrary, only 1 colony (2%) of the non-induced culture had the deletion genotype and there were no

colonies with mixed wild-type/deletion genotype retrieved (Supplementary Fig. 6). These results show that ThermoCas9 can be used as an efficient counter-selection tool in the mesophile *P. putida* KT2440 when grown at 37 °C.

**ThermoCas9-based gene silencing.** An efficient thermoactive transcriptional silencing CRISPRi tool is currently not available. Such a system could greatly facilitate metabolic studies of thermophiles. A catalytically dead variant of ThermoCas9 could serve this purpose by steadily binding to DNA elements without introducing dsDNA breaks. To this end, we identified the RuvC and HNH catalytic domains of ThermoCas9 and introduced the corresponding D8A and H582A mutations for creating a dead (d) ThermoCas9. After confirmation of the designed sequence, ThermodCas9 was heterologously produced, purified and used for an in vitro cleavage assay with the same DNA target as used in the aforementioned ThermoCas9 assays; no cleavage was observed confirming the catalytic inactivation of the nuclease.

Toward the development of a ThermodCas9-based CRISPRi tool, we aimed for the transcriptional silencing of the highly expressed *ldhL* gene from the genome of *B. smithii* ET 138. We constructed the pNW33n-based vectors pThermoCas9i_*ldh*L and pThermoCas9i_ctrl. Both vectors contained the *thermodCas9* gene under the control of $P_{xylL}$ promoter and a sgRNA expressing module under the control of the constitutive $P_{pta}$ promoter (Fig. 5c). The pThermoCas9i_*ldh*L plasmid contained a spacer for targeting the non-template DNA strand at the 5′ end of the 138 *ldh*L gene in *B. smithii* ET 138 (Supplementary Fig. 7). The position and targeted strand selection were based on previous studies[18,48], aiming for the efficient downregulation of the *ldhL* gene. The pThermoCas9i_ctrl plasmid contained a random non-targeting spacer in the sgRNA-expressing module. The constructs were used to transform *B. smithii* ET 138 competent cells at 55 °C followed by plating on LB2 agar plates, resulting in equal amounts of colonies. Two out of the ~700 colonies per construct were selected for culturing under microaerobic lactate-producing conditions for 24 h[44]. The growth of the pThermoCas9i_*ldh*L cultures was 50% less than the growth of the pThermoCas9i_ctrl cultures (Fig. 5d). We have previously shown that deletion of the *ldhL* gene leads to severe growth retardation in *B. smithii* ET 138 due to a lack of LDH-based NAD$^+$-regenerating capacity under microaerobic conditions[44]. Thus, the observed decrease in growth is likely caused by the transcriptional inhibition of the *ldhL* gene and subsequent redox imbalance due to loss of NAD$^+$-regenerating capacity. Indeed, HPLC analysis revealed 40% reduction in lactate production of the *ldhL* silenced cultures, and RT-qPCR analysis showed that the transcription levels of the *ldhL* gene were significantly reduced in the pThermoCas9i_*ldh*L cultures compared to the pThermoCas9i_ctrl cultures (Fig. 5d).

## Discussion

Most CRISPR-Cas applications are based on RNA-guided DNA interference by Class 2 CRISPR-Cas proteins, such as Cas9 and Cas12a[6–13]. Prior to this work, there were only a few examples of Class 1 CRISPR-Cas systems present in thermophilic bacteria and archaea[5,49], which have been used for genome editing of thermophiles[34]. As a result, the application of CRISPR-Cas technologies was mainly restricted to temperatures below 42 °C, due to the mesophilic nature of the employed Cas-endonucleases[28,29]. Hence, this has excluded application of these technologies in obligate thermophiles and in experimental approaches that require elevated temperatures and/or improved protein stability.

In the present study, we have characterized ThermoCas9, a Cas9 orthologue from the thermophilic bacterium *G. thermodenitrificans* T12 that has been isolated from compost[30]. Data

mining revealed additional Cas9 orthologues in the genomes of other thermophiles, which were nearly identical to ThermoCas9, showing that CRISPR-Cas type-II systems do exist in thermophiles, at least in some branches of the *Bacillus* and *Geobacillus* genera. We showed that ThermoCas9 is active in vitro in a wide temperature range of 20–70 °C, which is much broader than the 25–44 °C range of its mesophilic orthologue SpCas9. The extended activity and stability of ThermoCas9 allows for its application in molecular biology techniques that require DNA manipulation at temperatures of 20–70 °C, as well as its exploitation in harsh environments that require robust enzymatic activity. Furthermore, we identified several factors that are important for conferring the thermostability of ThermoCas9. First, we showed that the PAM preferences of ThermoCas9 are very strict for activity in the lower part of the temperature range (≤30 °C), whereas more variety in the PAM is allowed for activity at the moderate to optimal temperatures (37–60 °C). Second, we showed that ThermoCas9 activity and thermostability strongly depends on the association with an appropriate sgRNA guide. This stabilization of the multi-domain Cas9 protein is most likely the result of a major conformational change from an open/flexible state to a rather compact state, as described for SpCas9 upon guide binding[50]. Additionally, we showed that the ThermoCas9 activity on linear DNA targets is very sensitive to spacer-protospacer mismatches. At 55 °C, cleavage is affected of all linear fragments with single-mismatches ranging from position 1 (PAM proximal) to position 20 (PAM distal). Interestingly, positions 4, 5, and 10 are most important, whereas base pairing at position 2 appears less important. The same cleavage pattern is observed with plasmid targets. The elevated cleavage efficiencies of these targets suggest that the negatively supercoiled configuration of the used plasmids improves the target accessibility, as has been described before for the *E. coli* Type I-E CRISPR-Cas system[51]. Interestingly, despite overall lower activities, similar trends are observed in the case of cleavage assays of both linear fragments and plasmids with single-mismatches at 37 °C. The analysis of multiple mismatches reveals that the ThermoCas9 nuclease is extremely sensitive to four or more mismatches at the PAM distal end: at both temperatures cleavage is completely abolished of all tested targets. Even with two mismatches at positions 19–20, a severe drop of cleavage efficiency is observed. These results indicate a lower in vitro spacer-protospacer mismatch tolerance of ThermoCas9 compared to SpCas9[14] and highlight its potential as a genome editing tool for eukaryotic cells with enhanced target specificity. Elucidation of the ThermoCas9 crystal structure is required to gain insight on the molecular basis of the ThermoCas9 cleavage mechanism.

Based on the here described characterization of the novel ThermoCas9, we successfully developed genome engineering tools for strictly thermophilic prokaryotes. We showed that ThermoCas9 is active in vivo at 55 °C and 37 °C, and we adapted the current Cas9-based engineering technologies for the thermophile *B. smithii* ET 138 and the mesophile *P. putida* KT2440. Due to the wide temperature range of ThermoCas9, it is anticipated that the simple, effective and single plasmid-based ThermoCas9 approach will be suitable for a wide range of thermophilic and mesophilic microorganisms that can grow at temperatures from 37 °C up to 70 °C. This complements the existing mesophilic technologies, allowing their use for a large group of organisms for which these efficient tools were thus far unavailable.

Screening natural resources for novel enzymes with desired traits is unquestionably valuable. Previous studies have suggested that the adaptation of a mesophilic Cas9 orthologue to higher temperatures, with directed evolution and protein engineering, would be the best approach toward the construction of a

thermophilic Cas9 protein[34]. Instead, we identified a clade of Cas9 in some thermophilic bacteria, and transformed one of these thermostable ThermoCas9 variants into a powerful genome engineering tool for both thermophilic and mesophilic organisms. With this study, we further stretched the potential of the Cas9-based genome editing technologies and open new possibilities for using Cas9 technologies in novel applications under harsh conditions or requiring activity over a wide temperature range.

## Methods

**Bacterial strains and growth conditions.** The moderate thermophile *B. smithii* ET 138 Δ*sigF* Δ*hsdR*[28] was used for the gene editing and silencing experiments using ThermoCas9. It was grown in LB2 medium[44] at 55 °C. For plates, 30 g of agar (Difco) per liter of medium was used in all experiments. If needed chloramphenicol was added at the concentration of 7 μg mL⁻¹. For microaerobic lactate-producing conditions, *B. smithii* strains were grown in 25 mL TVMY medium[44] in a 50 mL Greiner tube for 24 h at 55 °C and 120 rpm after transfer from an overnight culture[44]. For protein expression, *E. coli* Rosetta (DE3) was grown in LB medium in flasks at 37 °C in a shaker incubator at 120 r.p.m. until an OD$_{600\ nm}$ of 0.5 was reached after the temperature was switched to 16 °C. After 30 min, expression was induced by addition of isopropyl-1-thio-β-D-gal-actopyranoside (IPTG) to a final concentration of 0.5 mM, after which incubation was continued at 16 °C. For cloning PAM constructs for sixth, and seventh, and eighth positions, DH5-alpha competent *E. coli* (NEB) was transformed according to the manual provided by the manufacturer and grown overnight on LB agar plates at 37 °C. For cloning degenerate 7-nt long PAM library, electro-competent DH10B *E. coli* cells were prepared and transformed using standard protocol[52] and grown on LB agar plates at 37 °C overnight. Chemically competent *E. coli* DH5α *λpir* (Invitrogen) was prepared and used for *P. putida* plasmid construction according to standard preparation and transformation protocols[52, 53]. For all *E. coli* strains, if required chloramphenicol was used in concentrations of 25 mg L⁻¹ and kanamycin in 50 mg L⁻¹. *P. putida* KT2440 (DSM 6125) strains were cultured at 37 °C in LB medium unless stated otherwise. If required, kanamycin was added in concentrations of 50 mg L⁻¹ and 3-methylbenzoate in a concentration of 3 mM.

**ThermoCas9 expression and purification.** ThermoCas9 was PCR amplified from the genome of *G. thermodenitrificans* T12, then cloned and heterologously expressed in *E. coli* Rosetta (DE3) and purified by FPLC by a combination of Ni²⁺-affinity, ion exchange and gel filtration chromatographic steps. The gene sequence was inserted into plasmid pML-1B (obtained from the UC Berkeley MacroLab, Addgene #29653) by ligation-independent cloning using oligonucleotides (Supplementary 2) to generate a protein expression construct encoding the ThermoCas9 polypeptide sequence (residues 1–1082) fused with an N-terminal tag comprising a hexahistidine sequence and a Tobacco Etch Virus (TEV) protease cleavage site. To express the catalytically inactive ThermoCas9 protein (ThermodCas9), the D8A and H582A point mutations were inserted using PCR and verified by DNA sequencing.

The proteins were expressed in *E. coli* Rosetta 2 (DE3) strain. Cultures were grown to an OD$_{600\ nm}$ of 0.5–0.6. Expression was induced by the addition of IPTG to a final concentration of 0.5 mM and incubation was continued at 16 °C overnight. Cells were collected by centrifugation and the cell pellet was resuspended in 20 mL of Lysis Buffer (50 mM sodium phosphate pH 8, 500 mM NaCl, 1 mM DTT, 10 mM imidazole) supplemented with protease inhibitors (Roche cOmplete, EDTA-free) and lysozyme. Once homogenized, cells were lysed by sonication (Sonoplus, Bandelin) using a using an ultrasonic MS72 microtip probe (Bandelin), for 5–8 min consisting of 2 s pulse and 2.5 s pause at 30% amplitude and then centrifuged at 16,000×g for 1 h at 4 °C to remove insoluble material. The clarified lysate was filtered through 0.22 micron filters (Mdi membrane technologies) and applied to a nickel column (Histrap HP, GE Lifesciences), washed and then eluted with 250 mM imidazole. Fractions containing ThermoCas9 were pooled and dialyzed overnight into the dialysis buffer (250 mM KCl, 20 mM HEPES/KOH, and 1 mM DTT, pH 8). After dialysis, sample was diluted 1:1 in 10 mM HEPES/KOH pH 8, and loaded on a heparin FF column pre-equilibrated in IEX-A buffer (150 mM KCl, 20 mM HEPES/KOH pH 8). Column was washed with IEX-A and then eluted with a gradient of IEX-C (2 M KCl, 20 mM HEPES/KOH pH 8). The sample was concentrated to 700 μL prior to loading on a gel filtration column (HiLoad 16/600 Superdex 200) via FPLC (AKTA Pure). Fractions from gel filtration were analyzed by SDS-PAGE; fractions containing ThermoCas9 were pooled and concentrated to 200 μL (50 mM sodium phosphate pH 8, 2 mM DTT, 5% glycerol, 500 mM NaCl) and either used directly for biochemical assays or frozen at −80 °C for storage.

**In vitro synthesis of sgRNA.** The sgRNA module was designed by fusing the predicted crRNA and tracrRNA sequences with a 5′-GAAA-3′ linker. The sgRNA-expressing DNA sequence was put under the transcriptional control of the T7 promoter. It was synthesized (Baseclear, Leiden, The Netherlands) and provided in the pUC57 backbone. All sgRNAs used in the biochemical reactions were synthesized using the HiScribe T7 High Yield RNA Synthesis Kit (NEB). PCR

fragments coding for sgRNAs, with the T7 sequence on the 5′ end, were utilized as templates for in vitro transcription reaction. T7 transcription was performed for 4 h. The sgRNAs were run and excised from urea-PAGE gels and purified using ethanol precipitation.

**In vitro cleavage assay.** In vitro cleavage assays were performed with purified recombinant ThermoCas9. ThermoCas9 protein, the in vitro transcribed sgRNA and the DNA substrates (generated using PCR amplification using primers described in Supplementary Table 2) were incubated separately (unless otherwise indicated) at the stated temperature for 10 min, followed by combining the components together and incubating them at the various assay temperatures in a cleavage buffer (100 mM sodium phosphate buffer (pH = 7), 500 mM NaCl, 25 mM MgCl₂, 25 (V/V%) glycerol, 5 mM dithiothreitol (DTT)) for 1 h. Each cleavage reaction contained 160 nM of ThermoCas9 protein, 4 nM of substrate DNA, and 150 nM of synthetized sgRNA. Reactions were stopped by adding 6X loading dye (NEB) and run on 1.5% agarose gels. Gels were stained with SYBR safe DNA stain (Life Technologies) and imaged with a Gel DocTM EZ gel imaging system (Bio-rad).

**Library construction for in vitro PAM screen.** For the construction of the PAM library, a 122-bp long DNA fragment, containing the protospacer and a 7-bp long degenerate sequence at its 3′-end, was constructed by primer annealing and Klenow fragment (exo-) (NEB) based extension. The PAM-library fragment and the pNW33n vector were digested by BspHI and BamHI (NEB) and then ligated (T4 ligase, NEB). The ligation mixture was transformed into electro-competent *E. coli* DH10B cells and plasmids were isolated from liquid cultures. For the 7 nt-long PAM determination process, the plasmid library was linearized by SapI (NEB) and used as the target. For the rest of the assays the DNA substrates were linearized by PCR amplification.

**PAM screening assay.** The PAM screening of thermoCas9 was performed using in vitro cleavage assays, which consisted of (per reaction): 160 nM of ThermoCas9, 150 nM in vitro transcribed sgRNA, 4 nM of DNA target, 4 μl of cleavage buffer (100 mM sodium phosphate buffer pH 7.5, 500 mM NaCl, 5 mM DTT, 25% glycerol) and MQ water up to 20 μl final reaction volume. The PAM containing cleavage fragments from the 55 °C reactions were gel purified, ligated with Illumina sequencing adaptors and sent for Illumina HiSeq 2500 sequencing (Baseclear). Equimolar amount of non-ThermoCas9 treated PAM library was subjected to the same process and sent for Illumina HiSeq 2500 sequencing as a reference. HiSeq reads with perfect sequence match to the reference sequence were selected for further analysis. From the selected reads, those present more than 1000 times in the ThermoCas9 treated library and at least 10 times more in the ThermoCas9 treated library compared to the control library were employed for WebLogo analysis[54].

**Library construction for in vitro mismatch tolerance screen.** For the construction of the spacer-protospacer mismatch target library, twenty pairs of 40-nt long complementary ssDNA fragments, containing the mismatch-protospacers, were annealed. The annealing products were designed to have overhangs compatible for their directional ligation (T4 ligase, NEB) into the pNW33n backbone, upon BspHI and BamHI (NEB) digestion of the vector. The ligation mixtures were transformed into chemically competent *E. coli* DH5α cells (NEB), plasmids were isolated from liquid cultures and verified by sequencing. Both plasmids and PCR-linearized DNA substrates were employed for the mismatch tolerance assays.

**B. smithii and P. putida editing and silencing constructs.** All the primers and plasmids used for plasmid construction were designed with appropriate overhangs for performing NEBuilder HiFi DNA assembly (NEB), and they are listed in Supplementary Tables 2, 3, respectively. The fragments for assembling the plasmids were obtained through PCR with Q5 Polymerase (NEB) or Phusion Flash High-Fidelity PCR Master Mix (ThermoFisher Scientific), the PCR products were subjected to 1% agarose gel electrophoresis and they were purified using Zymogen gel DNA recovery kit (Zymo Research). The assembled plasmids were transformed to chemically competent *E. coli* DH5α cells (NEB), or to *E. coli* DH5α λpir (Invitrogen) in the case of *P. putida* constructs, the latter to facilitate direct vector integration. Single colonies were inoculated in LB medium, plasmid material was isolated using the GeneJet plasmid miniprep kit (Thermo Fisher Scientific) and sequence verified (GATC-biotech) and 1 μg of each construct was transformed to *B. smithii* ET 138 electro-competent cells[44]. The MasterPure Gram Positive DNA Purification Kit (Epicentre) was used for genomic DNA isolation from *B. smithii* and *P. putida* liquid cultures. For the construction of the pThermoCas9_ctrl, pThermoCas9_bsΔpyrF1 and pThermoCas9_bsΔpyrF2 vectors, the pNW33n backbone together with the Δ*pyrF* homologous recombination flanks were PCR amplified from the pWUR_Cas9sp1_hr vector[28] (BG8191and BG8192). The native P$_{xylA}$ promoter was PCR amplified from the genome of *B. smithii* ET 138 (BG8194 and BG8195). The *thermocas9* gene was PCR amplified from the genome of *G. thermodenitrificans* T12 (BG8196 and BG8197). The P$_{pta}$ promoter was PCR amplified from the pWUR_Cas9sp1_hr vector[28] (BG8198 and BG8261_2/ BG8263_nc2/ BG8317_3). The spacers followed by the sgRNA scaffold were PCR

amplified from the pUC57_T7t12sgRNA vector (BG8266_2/BG8268_nc2/8320_3 and BG8210).

A four-fragment assembly was designed and executed for the construction of the pThermoCas9i_ldhL vectors. Initially, targeted point mutations were introduced to the codons of the *thermocas9* catalytic residues (mutations D8A and H582A), through a two-step PCR approach using pThermoCas9_ctrl as template. During the first PCR step (BG9075, BG9076), the desired mutations were introduced at the ends of the produced PCR fragment and during the second step (BG9091, BG9092) the produced fragment was employed as PCR template for the introduction of appropriate assembly-overhangs. The part of the *thermocas9* downstream the second mutation along with the *ldhL* silencing spacer was PCR amplified using pThermoCas9_ctrl as template (BG9077 and BG9267). The sgRNA scaffold together with the pNW33n backbone was PCR amplified using pThermoCas9_ctrl as template (BG9263 and BG9088). The promoter together with the part of the *thermocas9* upstream the first mutation was PCR amplified using pThermoCas9_ctrl as template (BG9089, BG9090).

A two-fragment assembly was designed and executed for the construction of pThermoCas9i_ctrl vector. The spacer sequence in the pThermoCas9i_ldhL vector was replaced with a random sequence containing BaeI restriction sites at both ends. The sgRNA scaffold together with the pNW33n backbone was PCR amplified using pThermoCas9_ctrl as template (BG9548, BG9601). The other half of the construct consisting of *thermodcas9* and promoter was amplified using pThermoCas9i_ldhL as template (BG9600, BG9549).

A five-fragment assembly was designed and executed for the construction of the *P. putida* KT2440 vector pThermoCas9_ppΔpyrF. The replicon from the suicide vector pEMG was PCR amplified (BG2365, BG2366). The flanking regions of *pyrF* were amplified from KT2440 genomic DNA (BG2367, BG2368 for the 576-bp upstream flank, and BG2369, BG2370 for the 540-bp downstream flank). The flanks were fused in an overlap extension PCR using primers BG2367 and BG2370 making use of the overlaps of primers BG2368 and BG2369. The sgRNA was amplified from the pThermoCas9_ctrl plasmid (BG2371, BG2372). The constitutive P3 promoter was amplified from pSW_I-SceI (BG2373, BG2374). This promoter fragment was fused to the sgRNA fragment in an overlap extension PCR using primers BG2372 and BG2373 making use of the overlaps of primers BG2371 and BG2374. ThermoCas9 was amplified from the pThermoCas9_ctrl plasmid (BG2375, BG2376). The inducible Pm-XylS system, to be used for 3-methylbenzoate induction of ThermoCas9 was amplified from pSW_I-SceI (BG2377, BG2378).

**Editing protocol for *P. putida*.** Transformation of the plasmid to electrocompetent *P. putida* was performed according to standard protocols[55]. After transformation and selection of integrants, overnight cultures were inoculated. 10 μl of overnight culture was used for inoculation of 3 ml fresh selective medium and after 2 h of growth at 37 °C ThermoCas9 was induced with 3-methylbenzoate. After an additional 6 h, dilutions of the culture were plated on non-selective medium supplemented with 3-methylbenzoate. For the control culture the addition of 3-methylbenzoate was omitted in all the steps. Confirmation of plasmid integration in the *P. putida* chromosome was done by colony PCR with primers BG2381 and BG2135. Confirmation of *pyrF* deletion was done by colony PCR with primers BG2381 and BG2382.

**RNA isolation.** RNA isolation was performed by the phenol as follows extraction based on a previously described protocol[56]: overnight 10 mL cultures were centrifuged at 4 °C and 4816×*g* for 15 min and immediately used for RNA isolation. After removal of the medium, cells were suspended in 0.5 mL of ice-cold TE buffer (pH 8.0) and kept on ice. All samples were divided into two 2 mL screw-capped tubes containing 0.5 g of zirconium beads, 30 μL of 10% SDS, 30 μL of 3 M sodium acetate (pH 5.2), and 500 μL of Roti-Phenol (pH 4.5–5.0, Carl Roth GmbH). Cells were disrupted using a FastPrep-24 apparatus (MP Biomedicals) at 5500 r.p.m. for 45 s and centrifuged at 4 °C and 9400×*g* for 5 min. 400 μL of the water phase from each tube was transferred to a new tube, to which 400 μL of chloroform–isoamyl alcohol (Carl Roth GmbH) was added, after which samples were centrifuged at 4 °C and 18,400×*g* for 3 min. 300 μL of the aqueous phase was transferred to a new tube and mixed with 300 μL of the lysis buffer from the high pure RNA isolation kit (Roche). Subsequently, the rest of the procedure from this kit was performed according to the manufacturer's protocol, except for the DNase incubation step, which was performed for 45 min. The concentration and integrity of cDNA was determined using Nanodrop-1000 Integrity and concentration of the isolated RNA was checked on a NanoDrop 1000.

**Quantification of mRNA by RT-qPCR.** First-strand cDNA synthesis was performed for the isolated RNA using SuperScript™ III Reverse Transcriptase (Invitrogen) according to manufacturer's protocol. qPCR was performed using the PerfeCTa SYBR Green Supermix for iQ from Quanta Biosciences. Quantity of 40 ng of each cDNA library was used as the template for qPCR. Two sets of primers were used; BG9665:BG9666 amplifying a 150-nt long region of the *ldhL* gene and BG9889:BG9890 amplifying a 150-nt long sequence of the *rpoD* (RNA polymerase sigma factor) gene which was used as the control for the qPCR. The qPCR was run on a Bio-Rad C1000 Thermal Cycler.

**High-pressure liquid chromatography.** A high-pressure liquid chromatography (HPLC) system ICS-5000 was used for lactate quantification. The system was operated with Aminex HPX 87H column from Bio-Rad Laboratories and equipped with a UV1000 detector operating on 210 nm and a RI-150 40 °C refractive index detector. The mobile phase consisted of 0.16 N $H_2SO_4$ and the column was operated at 0.8 mL min$^{-1}$. All samples were diluted 4:1 with 10 mM DMSO in 0.01 N $H_2SO_4$.

**Data availability.** Plasmids expressing ThermoCas9 or ThermodCas9, together with the corresponding sgRNA, are available on Addgene (#100981 & #100982).

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

## Acknowledgements

We would like to thank Koen Weenink, Steven Aalvink and Bastienne Vriesendorp for their technical assistance. R.v.K. and R.B.L.B. are employed by Corbion. I.M. and E.F.B. are supported by the Dutch Technology Foundation STW, which is part of The Netherlands Organization for Scientific Research (NWO) and which is partly funded by the Ministry of Economic Affairs. J.v.d.O. and P.M. are supported by the NWO/TOP grant 714.015.001.

## Author contributions

I.M., P.M., E.F.B., R.v.K., and J.v.d.O., conceived this study and design of experiments. I.M., P.M., E.F.B., M.F.B., V.V., M.I.S.N., A.G., and R.B.L.B. conducted the experiments. R.v.K. and J.v.d.O. supervised this project. I.M., P.M., E.F.B., R.v.K., and J.v.d.O. wrote the manuscript with input from all authors.

## Additional information

**Competing interests:** The authors declare no competing financial interests.

