## [Peer Review File · Nature Communications]

Reviewers' comments:

Reviewer #1 (Remarks to the Author):

Mougiakos characterize a thermostable CRISPR Type II-C Cas9 protein from *Geobacillus thermodenitrificans* T12 as a new genome editing tool. This work demonstrates ThermoCas9 is active at a range of temperatures up to 70°C which will facilitate many possible applications in a variety of thermophilic organisms. The authors characterized ThermoCas9 PAM variants, identified the tracrRNA, and have demonstrated a fused sgRNA will facilitate easier ThermoCas9 genome editing. The paper demonstrates editing in *Bacillus smithii*, a thermophile, and the mesophile *Pseudomonas putida*. The paper is well executed and will be broadly interesting to the genome engineering community as a new gene editing option for thermophiles.

The authors also demonstrate that ThermoCas9 can be mutated to create a Thermo-dCas9 or Thermo-CRISPRi system for control of transcription in thermophilic bacteria. To my knowledge this is the first implementation of transcriptional control using CRISPR in a thermophile and should be a useful tool in metabolic engineering in thermophiles or possibly in biotechnology applications.

Concerns:

Please make it more clear for a non-bacterial audience in lines 233-236 why a lack of colonies in pyrF1 and pyrF2 indicates active Cas9/sgRNA delivery that results in DSB that evidently if not repaired result in cell death in this strain of bacteria. In my understanding in *E. coli* if the break is not repaired by homologous repair then without NHEJ or other repair pathways chromosomal DSB result in cell death. I understand that the frequency of complete deletion is low in *Bacillus smithii* due to the low frequency of homologous repair off the DNA donor but it would make this section more convincing if the authors could use a plasmid assay or some non-chromosomal assay that does not convolute Cas9 editing activity with homologous repair to demonstrate how efficient the ThermoCas9 is at generating DNA DSBs in a thermophile.

For non-bacterial audiences it would broaden the impact or help if the authors make it clear why *Pseudomonas putida* and *Bacillus smithii* are interesting bacteria. At first pass it doesn't add much for the authors to use ThermoCas9 for editing in *Pseudomonas putida*. Why not use *Sp. Cas9*?

For both ThermoCas9 and Thermo-CRISPRi any demonstration of mismatch tolerance between the sgRNA and target DNA sequence would preliminarily characterize off target activity and suggest rules for avoiding off target activity at high and low temperatures. The PAM results at high and low temp for ThermoCas9 indicate the rules may be temperature dependent which is an interesting point.

Reviewer #2 (Remarks to the Author):

In this manuscript Mougiakos et al. characterize a thermo-stable Cas9 from the thermophilic bacterium *G. thermodenitrificans* and use it to perform gene deletion and gene silencing at high temperatures. The relevance of this work comes in great part from the fact that the commonly used Cas9 from *S. pyogenes* is not active at 42°C and higher as described here. The authors should however give proper credit to PMID: 27036863 for showing the thermosensitivity of *SpCas9*. The study was well conducted making several interesting observations, including how temperature affects PAM recognition and how tracrRNA hairpins enable activity at higher temperatures. The authors also convincingly show genome editing and gene silencing with Thermo-dCas9. Altogether these results provide interesting insights into a thermo-stable Cas9 and demonstrate its technological application which should be relevant to microbiologist working with

thermophilic bacteria.

Specific comments:

- The authors make the interesting observation that the sgRNA stabilizes Thermo-Cas9 at higher temperatures. I understand this is not the main objective of this technology driven paper, but it would be interesting to know how temperature affects the WT system with CRISPR and tracr. In particular does the processing step still occur, and is phage defense still effective at higher temperatures.
- Using the PAM-wheel representation proposed by Leenay et al. would give better insight than a sequence logo
- The description of the cation requirement would flow better with the rest of the manuscript if placed before other cleavage experiments are described. These results could also be better discussed. Can ThermoCas9 work with different cations? Is there a preferred metal ion? How does that compare to other described Cas9?

Rebuttal NCOMMS-17-14797

- Authors' responses are provided in blue.

Reviewer #1 (Remarks to the Author):

Mougiakos characterize a thermostable CRISPR Type II-C Cas9 protein from *Geobacillus thermodenitrificans* T12 as a new genome editing tool. This work demonstrates ThermoCas9 is active at a range of temperatures up to 70C which will facilitate many possible applications in a variety of thermophilic organisms. The authors characterized ThermoCas9 PAM variants, identified the tracrRNA, and have demonstrated a fused sgRNA will facilitate easier ThermoCas9 genome editing. The paper demonstrates editing in *Bacillus smithii*, a thermophile, and the mesophile *Pseudomonas putida*. The paper is well executed and will be broadly interesting to the genome engineering community as a new gene editing option for thermophiles.

The authors also demonstrate that ThermoCas9 can be mutated to create a Thermo-dCas9 or Thermo-CRISPRi system for control of transcription in thermophilic bacteria. To my knowledge this is the first implementation of transcriptional control using CRISPR in a thermophile and should be a useful tool in metabolic engineering in thermophiles or possibly in biotechnology applications.

- We thank Reviewer #1 for the kind words and the useful suggestions. Please find our replies to the specific concerns below.

Concerns:

1.1

Please make it more clear for a non-bacterial audience in lines 233-236 why a lack of colonies in *pyrF1* and *pyrF2* indicates active Cas9/sgRNA delivery that results in DSB that evidently if not repaired result in cell death in this strain of bacteria. In my understanding in *E coli* if the break is not repaired by homologous repair then without NHEJ or other repair pathways chromosomal DSB result in cell death. I understand that the frequency of complete deletion is low in *Bacillus smithii* due to the low frequency of homologous repair off the DNA donor but it would make this section more convincing if the authors could use a plasmid assay or some non-chromosomal assay that does not convolute Cas9 editing activity with homologous repair to demonstrate how efficient the ThermoCas9 is at generating DNA DSBs in a thermophile.

- This is indeed an important difference between eukaryotes and prokaryotes, which has big implications for the interpretation of the data. We agree with the reviewer that this should be described more elaborately, and hence we have re-written the text accordingly, as specified below. This extended explanation should also make it clear that there is no need for a separate plasmid assay that uses ThermoCas9 without HR, as we obtain zero colonies even in the presence of HR flanks. In our bacterial system this confirms the high *in vivo* activity of ThermoCas9 due to a combination of lacking NHEJ-type repair and poorly controllable ThermoCas9-expression.
- Updated text from line 242 onwards (additions to original text are in orange):

“In contrast, transformation with the pThermoCas9_bsΔpyrF1 and pThermoCas9_bsΔpyrF2 plasmids resulted in 20 and 0 colonies respectively. Out of 10 pThermoCas9_ΔpyrF1 colonies screened, 1 was a clean ΔpyrF mutant whereas the rest had a mixed wild type/ΔpyrF genotype (Figure 4B), proving the applicability of the system, as the designed homology directed repair of the targeted pyrF gene was successful. Contrary to eukaryotes, most prokaryotes including *B. smithii* do not possess a functional NHEJ system and DSBs induced by Cas9 have been shown to be lethal in the absence of an HR template or when cells do not successfully perform HR using this template^{11,28}. Hence, Cas9 functions as stringent counter-selection system to kill cells that have not performed the desired HR prior to Cas9 cleavage^{11,28,44}. The combination of lack of NHEJ and low HR-frequencies found in most prokaryotes forms the basis for the power of Cas9-based editing but also creates the need for tight control of Cas9 activity^{11,28,44}. As the promoter we use here for thermocas9-expression is not sufficiently controllable and HR is inefficient in *B. smithii*^{28,43}, the low number (pyrF1) or even complete lack (pyrF2) of colonies we observed here even in the presence of an HR template confirms the high *in vivo* activity of ThermoCas9 at 55°C. In the SpCas9-based counter-selection system we previously developed for *B. smithii*, the activity of Cas9 was very tightly controlled via the growth temperature instead of via promoter-controlled expression. This allowed for ample time for the cells to perform HR prior to Cas9 counter-selection, resulting in a higher pyrF deletion efficiency²⁸. We anticipate that the use of a tightly controllable promoter will increase efficiencies of the ThermoCas9-system.”

1.2

For non-bacterial audiences it would broaden the impact or help if the authors make it clear why *Pseudomonas putida* and *Bacillus smithii* are interesting bacteria. At first pass it doesn't add much for the authors to use ThermoCas9 for editing in *Pseudomonas putida*. Why not use Sp. Cas9?

- This is a good point and we have now extended the text to make this clearer for an audience not familiar with these organisms, as indicated below. We decided to work on editing *Pseudomonas putida* with ThermoCas9, since there is no published Cas9-based genome editing tool for this microbe³³ and colleagues from the *Pseudomonas* field confirmed that many attempts to engineer *Pseudomonas* strains with SpCas9 have not been successful (unpublished results). This fact, together with the increasing interest in this organism both as a fundamental model organism and as a biotechnological platform, made it a good model organism for our proof of principle mesophilic engineering.

- From line 224 onwards we added (in orange) about *B. smithii*:

“Its wide substrate utilization range, thermophilic and facultatively anaerobic nature, combined with its genetic amenability make this an organism with high potential as platform organism for the production of green chemicals in a biorefinery^{24,28,31,43}.”

- We have added the relevance of thermophiles from line 219 onwards:

“This group of bacteria is of great interest both from a fundamental as well as from an applied perspective. For biotechnological applications, their thermophilic nature results in for example less cooling costs, higher reaction rates and less contamination risk compared to the widely used mesophilic industrial work horses such as *E. coli*^{24,25,41,42}.”

- From line 264 onwards we have modified and added text about *P. putida*:

“To broaden the applicability of the ThermoCas9-based genome editing tool and to evaluate whether our *in vitro* results could be confirmed *in vivo*, we next evaluated its activity in the mesophilic Gram-negative bacterium *P. putida* KT2440. This soil bacterium is well-known for its unusual metabolism and biodegradation capacities, especially of aromatic compounds. Recently, interest in this organism has further increased due to its potential as platform host for biotechnology purposes using metabolic engineering^{45,46}. However, to

date no CRISPR-Cas9-based editing system has been reported for *P. putida* whereas such a system would greatly increase engineering efficiencies and enhance further study and use of this organism^{32,33}.”

- Additionally, we added (in orange) to line 88-89 in the introduction:
“In addition, we apply ThermoCas9 for *in vivo* genome editing of the mesophile *Pseudomonas putida* KT2440, for which to date no CRISPR-Cas9-based editing tool had been described^{32,33}, confirming the wide temperature range and broad applicability of this novel Cas9 system.”

1.3

For both ThermoCas9 and Thermo-CRISPRi any demonstration of mismatch tolerance between the sgRNA and target DNA sequence would preliminarily characterize off target activity and suggest rules for avoiding off target activity at high and low temperatures. The PAM results at high and low temp for ThermoCas9 indicate the rules may be temperature dependent which is an interesting point.

- Indeed off-target effects are an important issue in Cas9-editing. Since the focus of our work is on prokaryotes, which have small genomes (usually between 3 and 5 Mb), it is statistically easy to identify protospacers within almost every gene of prokaryotic genomes that contain the unique combination of seed and PAM sequences (approx. 8nt and 4nt long respectively). This way it is feasible to efficiently avoid the “off-target effects” problem that eukaryotic Cas9-based genome editing or silencing tools face. Moreover, most of the prokaryotes lack NHEJ, as also is our case, and any off-target effects will not result in wrongly edited cells but in cell death. If ThermoCas9 would induce dsDNA breaks in off-target sites – for which NO homologous ‘rescue’ templates are provided – the cells will not survive due to the lethality as discussed above (question/answer 1.1). In our case, in *B. smithii* ET 138 we observe cell death and reduced transformation efficiencies with ThermoCas9 even in the presence of HR templates, but as discussed above (1.1) this is most likely due to uncontrolled expression of ThermoCas9 before HR could take place, and does not reflect off-target effects. This is substantiated by the observation that we do not see this decrease in colonies in the editing process with *P. putida*, in which case we could make use of a tightly controllable promoter. Further research is needed to evaluate this in more detail, but this would be a major side-project that we consider out of the scope of the present paper. However, we agree this would be an interesting follow-up project, for instance in combination with a study on the mentioned effects of the temperature on ThermoCas9 functionality (see also reviewer 2.1).

Reviewer #2 (Remarks to the Author):

In this manuscript Mougiakos et al. characterize a thermos-stable Cas9 from the thermophilic bacterium *G. thermodenitrificans* and use it to perform gene deletion and gene silencing at high temperatures. The relevance of this work comes in great part from the fact that the commonly used Cas9 from *S. pyogenes* is not active at 42°C and higher as described here. The authors should however give proper credit to PMID: 27036863 for showing the thermosensitivity of SpCas9. The study was well conducted making several interesting observations, including how temperature affects PAM recognition and how tracrRNA hairpins enable activity at higher temperatures. The authors also convincingly show genome editing and gene silencing with Thermo-dCas9. Altogether these results provide interesting insights into a thermo-stable Cas9 and demonstrate its technological application which should be relevant to microbiologist working with thermophilic bacteria.

- We thank Reviewer #2 for the kind words and the useful suggestions. Please find our replies to the specific concerns below. As for the above comment on credit to PMID 27036863, we thank the reviewer for making us aware of this highly interesting paper and we have inserted references to it in lines 79, 209 and 329.

Specific comments:

2.1

The authors make the interesting observation that the sgRNA stabilizes Thermo-Cas9 at higher temperatures. I understand this is not the main objective of this technology driven paper, but it would be interesting to know how temperature affects the WT system with CRISPR and tracr. In particular does the processing step still occur, and is phage defense still effective at higher temperatures.

- We agree with the reviewer that this would very interesting to know. But as also mentioned by the reviewer, it is out of the scope of this paper. We would like to make this part of the future study also mentioned above in reply to question 1.3, to look further into the effects of the temperature on ThermoCas9 functionality in a more fundamental study.

2.2

Using the PAM-wheel representation proposed by Leenay et al. would give better insight than a sequence logo

- We agree with the reviewer that the PAM-wheel could indeed give an overall insight of the sequencing results obtained from the initial PAM determination process that exhibited the high importance of the 5th PAM position. However, we would also like to bring to focus that the final PAM identification was based on an experimental approach that did not result in sequencing data as output, making the construction of a PAM-wheel not feasible. A series of extensive PAM assays were executed in a step-wise manner, in order to obtain with the highest possible detail the preferences for the 6th, 7th and 8th PAM positions at different temperatures. Therefore, in this scenario we decided to use the sequence logo as

an estimation of the first 7 positions of the PAM and the *in vitro* cleavage assays for the final selection of the PAM, since we cannot display these different experiments in a single PAM-wheel (or in a single Weblogo).

2.3

The description of the cation requirement would flow better with the rest of the manuscript if placed before other cleavage experiments are described. These results could also be better discussed. Can ThermoCas9 work with different cations? Is there a preferred metal ion? How does that compare to other described Cas9?

- This is a good suggestion. Hence, we have now placed the cation requirement analysis before other cleavage experiments (thermostability and truncations) in order to improve the flow of the text.
- We have also added the following sentences (lines 165-174) in order to further describe the ThermoCas9 activity with different cations, the ThermoCas9 preferred metal ions and how they compare with other described Cas9s:

“Previously characterized, mesophilic Cas9 endonucleases employ divalent cations to catalyze the generation of DSBs in target DNA^{14,37}. To determine the ion dependency of ThermoCas9 cleavage activity, plasmid cleavage assays were performed in the presence of one of the following divalent cations: Mg²⁺, Ca²⁺, Mn²⁺, Fe²⁺, Co²⁺, Ni²⁺, and Zn²⁺; an assay with the cation-chelating agent EDTA was included as negative control. As expected, target dsDNA was cleaved in the presence of divalent cations and remained intact in the presence of EDTA (Supplementary Figure 5A). The DNA cleavage activity of ThermoCas9 was the highest when Mg²⁺ and Mn²⁺ was added to the reaction consistent with other Cas9 variants^{14,20,39}. Addition of Fe²⁺, Co²⁺, Ni²⁺, or Zn²⁺ ions also mediated cleavage. Ca²⁺ only supported plasmid nicking, suggesting that with this cation only one of the endonuclease domains is functional.”

REVIEWERS' COMMENTS:

Reviewer #1 (Remarks to the Author):

this paper will be widely interesting and useful and should be published in Nature Communications. the authors respond to both reviews with substantial text edits but no experiments which would define the mismatch tolerance for either PAM or protospacer mismatches. This idea was raised by both reviewers as very useful characterization of the system and in my mind is not outside the scope of the paper. A simple experiment could be to express 3-10 sgRNAs with mismatches in the protospacer and measure CRISPR or CRISPRi activity at low and high temperatures. I agree for CRISPR that off target activity in their system will not be a problem as the cells will die but for CRISPRi this could still produce off target transcription changes.

At minimum the authors should clearly state that off target activity is not a problem in their systems as the bacteria will die but must be tested for other systems that may have other DNA repair properties.

Reviewer #2 (Remarks to the Author):

The points I raised have been satisfactorily addressed.

Rebuttal 2nd revision NCOMMS-17-14797

Please find our responses to the reviewers in blue.

REVIEWERS' COMMENTS:

Reviewer #1 (Remarks to the Author):

this paper will be widely interesting and useful and should be published in Nature Communications. the authors respond to both reviews with substantial text edits but no experiments which would define the mismatch tolerance for either PAM or protospacer mismatches. This idea was raised by both reviewers as very useful characterization of the system and in my mind is not outside the scope of the paper. A simple experiment could be to express 3-10 sgRNAs with mismatches in the protospacer and measure CRISPR or CRISPRi activity at low and high temperatures. I agree for CRISPR that off target activity in their system will not be a problem as the cells will die but for CRISPRi this could still produce off target transcription changes.

At minimum the authors should clearly state that off target activity is not a problem in their systems as the bacteria will die but must be tested for other systems that may have other DNA repair properties.

We thank the reviewer for his/her positive words about our work and for suggesting this useful experiment. We designed and constructed a series of 20 protospacer containing plasmids harboring single or multiple mismatches compared to the spacer of the employed sgRNA. We have performed *in vitro* assays at 37°C and 55°C using these plasmids and their linearized PCR products as DNA targets. The results of this experiment are added in the results section of the paper under the subheading “Spacer-protospacer mismatch tolerance” between lines 218 and 242. The discussion of the results is added in the discussion section between lines 375 and 392. The method we used to construct the DNA targets for this experiment was added in the methods section under the subheading “Library construction for *in vitro* mismatch tolerance screen” between lines 509 and 517. Finally, the results are now depicted in a separate figure, figure 4. The legend of this figure is added between lines 840 and 855.

Attempts to perform *in vivo* analyses of the mismatch constructs at elevated temperatures were inconclusive for the different single mismatches tested, due to highly variable transformation efficiencies of *Bacillus smithii* (not shown). The targeting construct though constantly gave much less number of surviving colonies as compared to the mismatch constructs.

Reviewer #2 (Remarks to the Author):

The points I raised have been satisfactorily addressed.